# Upregulation of ERK phosphorylation in rat dorsal root ganglion neurons contributes to oxaliplatin-induced chronic neuropathic pain

**Toyoaki Maruta**[1]☯*, **Takayuki Nemoto**[2]☯, **Koutaro Hidaka**[1], **Tomohiro Koshida**[1], **Tetsuro Shirasaka**[1], **Toshihiko Yanagita**[3], **Ryu Takeya**[4], **Isao Tsuneyoshi**[1]

**1** Department of Anesthesiology, Faculty of Medicine, University of Miyazaki, Miyazaki, Japan, **2** Department of Pharmacology, Faculty of Medicine, Fukuoka University, Fukuoka, Japan, **3** Department of Clinical Pharmacology, School of Nursing, Faculty of Medicine, University of Miyazaki, Miyazaki, Japan, **4** Department of Pharmacology, Faculty of Medicine, University of Miyazaki, Miyazaki, Japan

☯ These authors contributed equally to this work.
* mmctm2@yahoo.co.jp

**Data Availability Statement:** All relevant data are within the paper and its Supporting Information files.

## Abstract

Oxaliplatin is the first-line chemotherapy for metastatic colorectal cancer. Unlike other platinum anticancer agents, oxaliplatin does not result in significant renal impairment and ototoxicity. Oxaliplatin, however, has been associated with acute and chronic peripheral neuropathies. Despite the awareness of these side-effects, the underlying mechanisms are yet to be clearly established. Therefore, in this study, we aimed to understand the factors involved in the generation of chronic neuropathy elicited by oxaliplatin treatment. We established a rat model of oxaliplatin-induced neuropathic pain (4 mg kg$^{-1}$ intraperitoneally). The paw withdrawal thresholds were assessed at different time-points after the treatment, and a significant decrease was observed 3 and 4 weeks after oxaliplatin treatment as compared to the vehicle treatment (4.4 ± 1.0 vs. 16.0 ± 4.1 g; $P$ < 0.05 and 4.4 ± 0.7 vs. 14.8 ± 3.1 g; $P$ < 0.05, respectively). We further evaluated the role of different mitogen-activated protein kinases (MAPKs) pathways in the pathophysiology of neuropathic pain. Although the levels of total extracellular signal-regulated kinase (ERK) 1/2 in the dorsal root ganglia (DRG) were not different between oxaliplatin and vehicle treatment groups, phosphorylated ERK (p-ERK) 1/2 was up-regulated up to 4.5-fold in the oxaliplatin group. Administration of ERK inhibitor PD98059 (6 μg day$^{-1}$ intrathecally) inhibited oxaliplatin-induced ERK phosphorylation and neuropathic pain. Therefore, upregulation of p-ERK by oxaliplatin in rat DRG and inhibition of mechanical allodynia by an ERK inhibitor in the present study may provide a better understanding of intracellular molecular alterations associated with oxaliplatin-induced neuropathic pain and help in the development of potential therapeutics.

## Introduction

Oxaliplatin, a platinum-based drug, is used as the first-line chemotherapy for metastatic colorectal cancer. Unlike other platinum anticancer agents, oxaliplatin does not result in

**Funding:** This study was supported by a Grant for Clinical Research from University of Miyazaki Hospital. The funders had no role in study design, data collection and analysis, decision to publish, or preparation of the manuscript.

**Competing interests:** The authors have declared that no competing interests exist.

significant renal impairment and ototoxicity. However, oxaliplatin is associated with acute and chronic peripheral neuropathies [1, 2]. Oxaliplatin-induced acute neuropathy is characterized by acral paresthesia that is enhanced by exposure to cold. Furthermore, cumulative oxaliplatin dose can cause chronic neuropathy, which includes pain, paresthesia, hypoesthesia, dysesthesia, and changes in proprioception. Therefore, oxaliplatin-induced neuropathic pain is a major clinical side-effect that can influence the treatment as well as the quality of life.

Pain results from the activation of a subset of sensory neurons termed nociceptors. Under physiological conditions, activation of unmyelinated (C-fiber) and myelinated (Aδ-fiber) nociceptive afferent fibers indicates potential tissue damage, which is reflected in the high thresholds of nociceptors for mechanical, thermal, and chemical stimuli; these neurotransmissions are attributed to ion channels, neurotransmitters, and intracellular signaling [3, 4]. These conditions change dramatically in neuropathic pain states, including chemotherapy-induced peripheral neuropathy (CIPN). Understanding the changes that occur in neuropathic pain is vital to identify new therapeutic targets and develop novel analgesics [4]. Recently, it has been reported that oxaliplatin-induced acute paresthesia is induced by voltage-dependent sodium channel ($Na_V1.6$) dysfunction [5–7] and upregulation of transient receptor potential (TRP) channels, TRPM8 and TRPA1 [8–11], which are temperature-sensitive channels. However, the pathophysiology of oxaliplatin-induced neuropathic pain as a chronic neuropathy has not yet been clearly established.

Mitogen-activated protein kinases (MAPKs) signaling cascade is known to be involved in the regulation of cellular functions such as cell differentiation, proliferation, and apoptosis [12, 13]. MAPKs, such as extracellular signal-regulated kinase (ERK), p38 kinase, and c-jun N-terminal kinase (JNK), have been linked with the development of pain [12, 13]. Furthermore, it has recently been reported that the modulation of MAPKs activation is associated with oxaliplatin-induced apoptosis in cultured dorsal root ganglion (DRG) neurons [14, 15].

Therefore, the aim of the current study was to understand the factors involved in the generation of chronic neuropathy elicited by oxaliplatin treatment. We investigated whether MAPKs were modulated by oxaliplatin in the rat DRG and found that oxaliplatin treatment up-regulates ERK phosphorylation in rat DRG and induced chronic neuropathic pain. We also demonstrated that administration of an ERK inhibitor inhibits oxaliplatin-induced neuropathic pain. Thus, our study suggests a novel mechanism by which oxaliplatin treatment can influence MAPKs signaling and contribute to chronic neuropathy.

## Materials and methods

### Animals

Six-week-old male Sprague Dawley rats (Kudo, Japan) weighing approximately 200–250 g were used in the study. All rats were individually housed in a temperature- and humidity-controlled environment with a 12-hour light-dark cycle and were permitted free access to food and water. The study was conducted in strict accordance with the guidelines for Proper Conduct of Animal Experiments (Science Council of Japan). The experiments were approved by the Experimental Animal Care and Use Committee of University of Miyazaki (Permit Number: 2015–528). All efforts were made to minimize the number of animals used and their suffering.

### Pharmacological treatments

In the first series of experiments, we investigated the intracellular molecular alterations in DRG. Oxaliplatin (4 mg kg$^{-1}$ of body weight; Sigma-Aldrich, St. Louis, MO, USA) or vehicle (5% glucose) was injected intraperitoneally (i.p.) twice a week for 4 weeks [16]. Oxaliplatin was

prepared in 5% glucose to a final concentration of 2 mg ml$^{-1}$. von Frey test was conducted before and 1 week after each oxaliplatin or vehicle treatment. On day 28, at the end of the last behavioral test, L4-L6 DRGs were dissected from each group and intracellular molecules including ERK were measured using western blot analysis.

In the second series of experiments, we investigated the effects of ERK inhibitor on oxaliplatin-induced neuropathy. Rats were anesthetized with an intraperitoneal injection of a combination anesthetic (0.375 mg kg$^{-1}$ of medetomidine, 2.0 mg kg$^{-1}$ of midazolam, and 2.5 mg kg$^{-1}$ of butorphanol). A PE-10 polyethylene catheter (Becton-Dickinson, Sparks, MD, USA) was inserted into the subarachnoid space through the atlanto-occipital membrane and pushed to the region of lumbar enlargement [17]. An osmotic pressure pump (ALZET model 2004, DURECT, Cupertino, CA, USA; total volume of the pump: 200 µl, drug infusion: 0.25 µl hour$^{-1}$ for 4 weeks) was connected to the catheter and placed subcutaneously on the back [18]. Immediately after surgery, the operating surgeon regularly observed animals until they were ambulatory. Furthermore, the animals' appearance, movement, and appetite were observed daily for one week after surgery. ERK inhibitor PD98059 (6 µg day$^{-1}$; Sigma-Aldrich, St. Louis, MO, USA) or vehicle [20% dimethyl sulfoxide (DMSO)] was injected intrathecally (i.t.) for 4 weeks with the osmotic pressure pump. PD98059 was dissolved in 20% DMSO to a final concentration of 1 µg µl$^{-1}$. One week after the pump placement, oxaliplatin (4 mg kg$^{-1}$) or vehicle (5% glucose) was injected i.p. twice a week for 3 weeks. The sham-operated (Sham) mice underwent a similar surgical procedure except for pump placement and drug treatments. von Frey test was conducted before pump placement, and before and 1 week after each oxaliplatin or vehicle treatment. On day 28 of pump placement, at the end of the last behavioral test, L4-L6 DRGs were dissected from each group and intracellular levels of various molecules were measured using western blot analysis.

## von Frey test

Mechanical sensitivity was examined by testing the paw withdrawal threshold using the von Frey (VF) filaments (Stoelting, Wood Dale, IL, USA). Briefly, each rat was placed in a 20 cm × 20 cm suspended chamber on a metallic mesh floor. After an acclimation period of 30 minutes, a series of calibrated VF filaments were applied perpendicularly to the plantar surface of the right and left hind paws with sufficient force to bend the filament for 5 seconds. Brisk withdrawal or paw flinching was considered as a positive response. In the absence of a response, the filament of next greater force was applied. After a positive response, the filament of next lower force was applied. The tactile stimulus producing a 50% likelihood of withdrawal response was calculated using the up-down method [19]. von Frey test was conducted before an osmotic pressure pump placement, and before and 1 week after each intraperitoneal oxaliplatin or vehicle treatment for 4 weeks.

## Western blot analysis

On day 28 of intraperitoneal treatment or pump placement, all rats were euthanized with sevoflurane exposure and the L4-L6 DRGs from each group were quickly dissected for further analysis. Briefly, the collected DRGs were mechanically homogenized in ice-cold lysis buffer composed of 1% Triton X-100, 150 mM NaCl, 1 mM EDTA, and 20 mM Tris-HCl, pH 7.5, with added Protease and Phosphatase Inhibitor Cocktails (Roche Diagnostics, Mannheim, Germany) and centrifuged at 12000 rpm and 4°C for 10 minutes. The supernatant was collected and stored at -80°C until use. The total protein content was determined in each sample using the Bradford method-based protein assay kit, with bovine serum albumin (BSA) as standard (Aproscience, Naruto, Japan). The supernatants were solubilized in 2× SDS

electrophoresis sample buffer and heated at 98°C for 5 minutes. Equal amount of proteins (7.0–7.5 μg per lane) were separated by SDS-12% polyacrylamide gel electrophoresis (PAGE) and transferred onto a polyvinylidene difluoride (PVDF) membrane (Immobilon-P, Merck Millipore, Burlington, MA, USA). The membrane was then incubated with a blocking solution [5% BSA in Tween-Tris-buffered saline (10 mM Tris-HCl, pH 7.4, 150 mM NaCl, and 0.1% Tween-20)] and further incubated overnight at 4°C in Can Get Signal Solution-1 (TOYOBO, Osaka, Japan) with rabbit anti-ERK polyclonal antibody (1:2000, K-23, Santa Cruz, Dallas, TX, USA), mouse anti-p-ERK monoclonal antibody (1:2000, E-4, Santa Cruz), rabbit anti-p38 monoclonal antibody (1:2000, D13E1, Cell Signaling Technology, MA, USA), rabbit anti-p-p38 monoclonal antibody (Thr180/Tyr182) (1:2000, D3F9, Cell Signaling Technology), rabbit anti-JNK polyclonal antibody (1:2000, D-2, Santa Cruz), mouse anti-p-JNK monoclonal antibody (1:2000, Santa Cruz), rabbit anti-BDNF polyclonal antibody (1:2000, ab226843, Abcam, Cambridge, UK) or mouse anti-β-actin monoclonal antibody (1:2000, A1978, Sigma-Aldrich, St. Louis, MO, USA). After repeated washing, the immunoreactive bands were developed using Can Get Signal Solution-2 with horseradish peroxidase-conjugated anti-rabbit antibody (1:5000, GE Healthcare Japan Corporation, Tokyo, Japan) or anti-mouse antibody (1:5000, Santa Cruz), then visualized using an enhanced chemiluminescence detection system reagent (Amersham ECL-prime, GE Healthcare Japan Corporation), and captured in a LAS-3000 Luminoimage analyzer (Fuji Film, Tokyo, Japan). A commercially available molecular weight marker (Amersham ECL rainbow marker–full range, GE Healthcare Japan Corporation), consisting of proteins of molecular weight 12 to 225 kDa, was used as a reference for each molecular weight. The densities of protein blots were quantified using ImageJ [20] and the protein levels were normalized to β-actin levels.

## Statistical analysis

For behavioral experiments, the hindpaw data within each group were analyzed using one-way repeated measures analysis of variance (ANOVA) followed by Bonferroni post hoc analysis. Comparisons between two means of the hindpaw data and western blot data were performed by Welch's test and Student's t-test, respectively. The results were presented as mean ± SEM (for von Frey test) or ± SD (for western blot analysis). $P < 0.05$ was considered as significant. The statistics software used was JMP 11 (SAS Institute, Inc., Cary, NC, USA) for Macintosh.

# Results

## Mechanical allodynia in a rat model of oxaliplatin-induced neuropathic pain

The oxaliplatin treatment (4 mg kg$^{-1}$, twice a week for 4 weeks) induced increased pain behavior in the rat model [16]. Fig 1 shows that the paw withdrawal thresholds measured with VF filaments to the non-noxious mechanical stimulus at 3 and 4 weeks after oxaliplatin treatment were significantly lower than the vehicle treatment (4.4 ± 1.0 g vs. 16.0 ± 4.1 g; $P = 0.046$ and 4.4 ± 0.7 g vs. 14.8 ± 3.1 g; $P = 0.027$, respectively).

## Oxaliplatin treatment up-regulates ERK phosphorylation in rat DRG neurons

The modulation of MAPKs activation, including ERK, p38 kinase, and JNK pathways, have not only been linked with the development of pain [12, 13], but also with the oxaliplatin-induced apoptosis in DRG [14, 15]. Western blot analyses of different MAPKs in the DRG of oxaliplatin-treated rats as compared to vehicle-treated rats are illustrated in Figs 2–4. Although

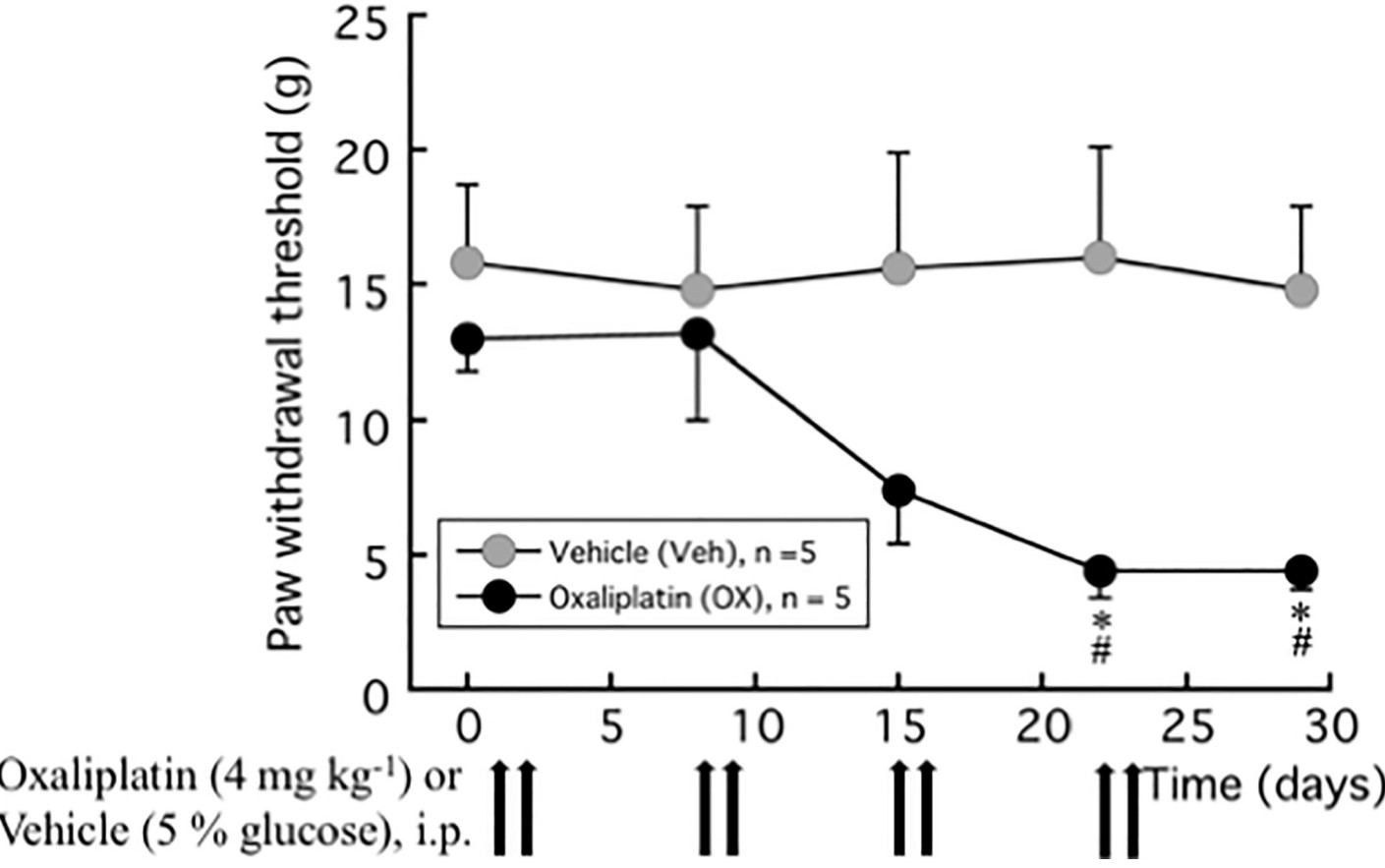

**Fig 1. Paw withdrawal test (von Frey test) for mechanical allodynia induced by oxaliplatin.** Oxaliplatin (4 mg kg$^{-1}$) was administered i.p. twice a week for 4 weeks (days 1, 2, 8, 9, 15, 16, 22, and 23). We confirmed the incidence of mechanical allodynia on day 28. The von Frey test was performed before and 1 week after each oxaliplatin or vehicle (5% glucose) treatment. The hindpaw data within each group were analyzed using one-way repeated measures ANOVA followed by Bonferroni post hoc analysis. For comparisons between groups at the same time, Welch's test was used. All data are calculated as mean ± SEM of 5 animals. * $P < 0.05$, compared with Time 0 (baseline). # $P < 0.05$, compared with the vehicle at the same time.

no difference was observed in the total ERK1/2 levels between oxaliplatin and vehicle treatment groups, p-ERK1/2 was found to be up-regulated up to 4.5-fold (447.6 ± 273.6%, $P = 0.0029$) in DRG of oxaliplatin-induced neuropathic pain rat model (Fig 2). On the other hand, no change was observed in the phosphorylation and protein levels of p38 and JNK between oxaliplatin and vehicle treatment groups (Figs 3 and 4).

## Oxaliplatin treatment increases brain-derived neurotrophic factor (BDNF) levels in rat DRG

BDNF is not only a nerve growth factor, but also a neurotransmitter of nociceptive fibers in the dorsal horn of the spinal cord. In the spinal nerve ligation (SNL) model of neuropathic pain, BDNF expression was found to be up-regulated in the rat spinal dorsal horn [21]. BDNF expression was also up-regulated in DRG in lumbar 5 ventral root transection model of neuropathic pain [21].

In our study, BDNF levels were increased in DRG of oxaliplatin-induced neuropathic pain rat model (115.8 ± 23.4%, $P = 0.047$) (Fig 5).

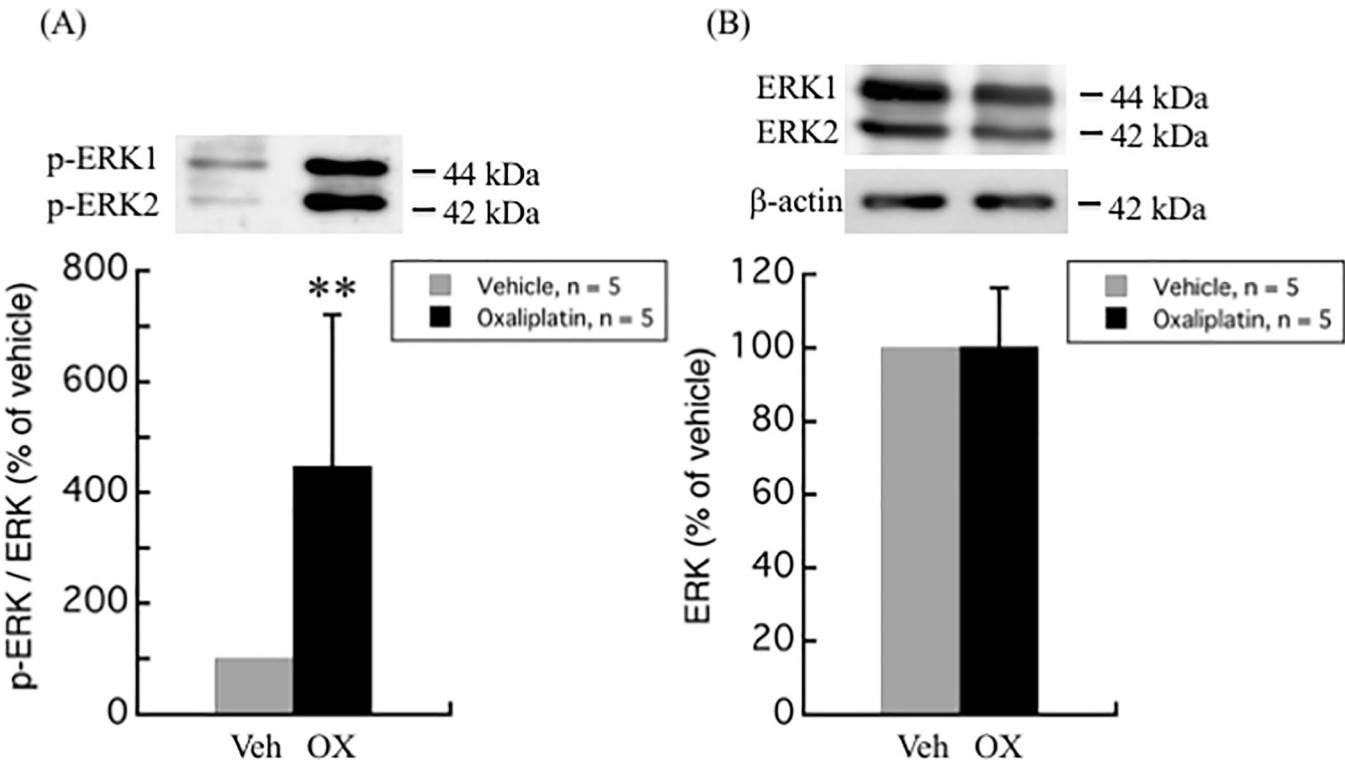

**Fig 2. Upregulation of ERK phosphorylation by oxaliplatin in rat DRG.** (A) The ratio of p-ERK to ERK expression was significantly increased in DRG of oxaliplatin treated rats. (B) No difference was observed in the protein levels of ERK between oxaliplatin and vehicle treatment groups. Comparisons between two groups of the blots were performed by Student's t-test. All data are calculated as mean ± SD of 5 animals. ** $P < 0.001$, compared to the vehicle.

## Effects of ERK inhibitor on oxaliplatin-induced neuropathic pain

Fig 6 shows that ERK inhibitor PD98059 (6 μg day$^{-1}$) injected intrathecally inhibited oxaliplatin-induced mechanical allodynia. The paw withdrawal thresholds in the oxaliplatin and PD98059 treatment group were mostly maintained from baseline and were significantly higher than the oxaliplatin and vehicle treatment group at 3 weeks after oxaliplatin treatment (OX + PD98059: 15.2 ± 2.9 g and Sham: 16.2 ± 2.6 g vs. OX + Veh: 4.8 ± 0.8 g; $P = 0.021$ and $P = 0.01$, respectively). Concomitantly, PD98059 also inhibited oxaliplatin-induced upregulation of ERK phosphorylation in DRG (OX + PD98059: 147.6 ± 47.6%, vs. OX + Veh: 402.9 ± 251.2%, $P = 0.0094$) (Fig 7).

## Discussion

Platinum-based drugs are the first-line chemotherapy for different cancers. Platinum derivatives such as oxaliplatin and cisplatin act as cytotoxins on tumor cells by forming platinum-DNA adducts, thus leading the tumor cells to programmed cell death. These platinum derivatives induce Chemotherapy-Induced Peripheral Neuropathy (CIPN) as one of the clinical side-effects [1, 2]. In an *in-vitro* study, treatment of cultured DRG neurons from E15 rat embryos with toxic doses of oxaliplatin or cisplatin induced a dose-dependent neuronal apoptosis by phosphorylating and inactivating the anti-apoptotic protein Bcl-2 and increasing the levels of the pro-apoptotic protein Bax [14].

Furthermore, studies have shown that these platinum derivatives modulate different MAPKs [13]. MAPKs are vital for intracellular signal transduction and play critical roles in

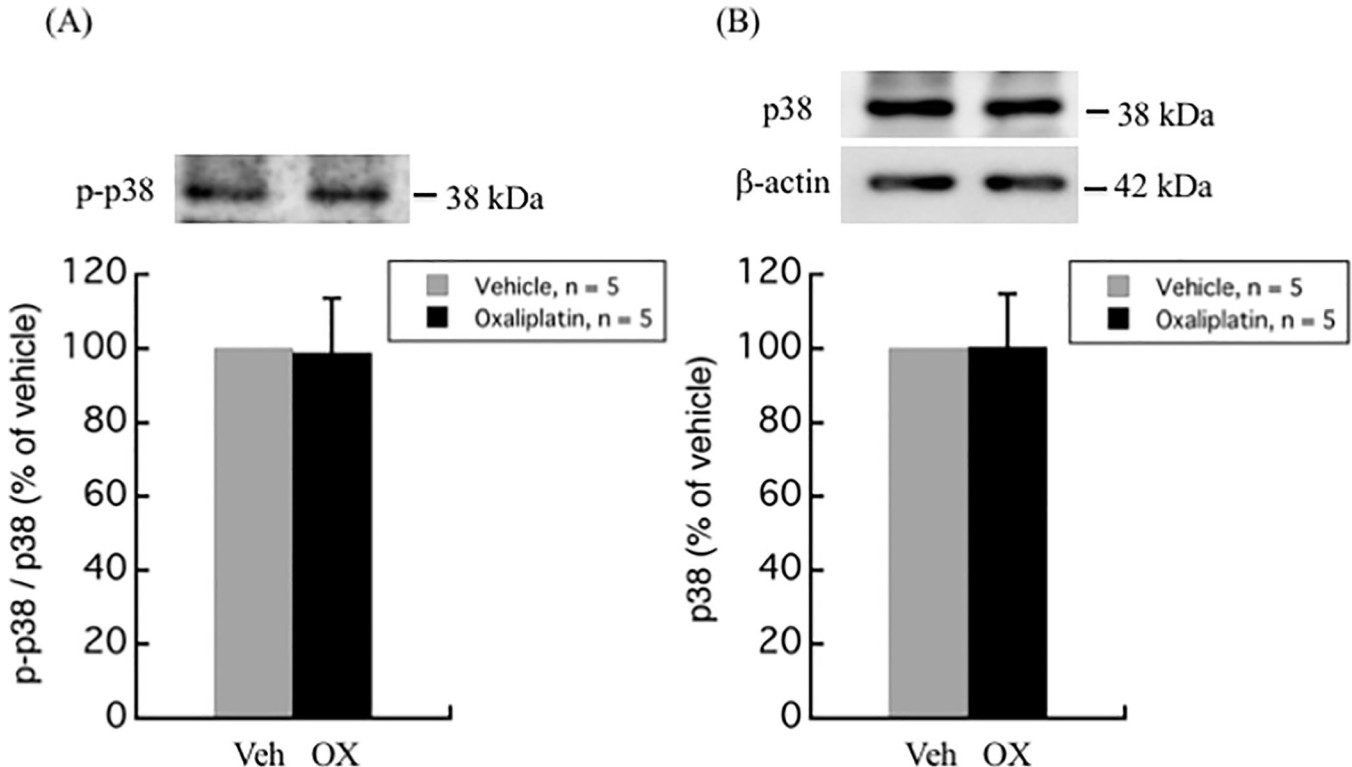

**Fig 3. p38 phosphorylation and protein levels in rat DRG.** There was no difference in p38 phosphorylation and protein levels between oxaliplatin and vehicle treatment groups. Comparisons between two groups of the blots were performed by Student's t-test. All data are calculated as mean ± SD of 5 animals.

regulation of neural plasticity and inflammatory responses [12, 13]. This family of kinases consists of three key members: ERK, p38, and JNK. Accumulating evidence shows that the activation of MAPKs can induce the synthesis of pronociceptive mediators via distinct molecular and cellular mechanisms, resulting in the enhancement and prolongation of pain [12, 13]. The platinum derivatives phosphorylate and activate p38 while they reduce the levels of active and total JNK. Both oxaliplatin and cisplatin have shown to activate ERKs during early stages (4–8 hours after treatment), although they behave differently at later stages [14]. Moreover, by using specific inhibitors of the different MAPKs, it has been demonstrated that the platinum-induced neuronal apoptosis is mediated by early p38 and ERK1/2 activation [14]. In *in-vivo* studies, oxaliplatin has shown to increase p38 phosphorylation at 0.5 and 4 hours after the treatment [22], or protein kinase C (PKC) phosphorylation, ERK1/2 phosphorylation, and c-fos expression on day 14 after the treatment [23], in the spinal cord of oxaliplatin-induced neuropathy mouse model. These results and our findings suggest a role for MAPKs including ERK in the generation and development of oxaliplatin-induced peripheral neuropathy.

Electrophysiological studies in patients undergoing oxaliplatin-treatment demonstrated nerve hyperexcitability in both peripheral motor [24] and sensory axons [25]. Furthermore, electrophysiological *in-vitro* studies in isolated peripheral nerve segments indicated that the hyperexcitability is characterized by an increase in the duration of the compound A-fiber action potential and the emergence of after-activity persisting over several tens of milliseconds [26–29]. A modulating effect on both voltage-dependent sodium channels and delayed rectifier potassium channels has been demonstrated during oxaliplatin administration to the myelinated axons in frog nerves [30] and to the neuronal cells in cell culture [31]. Thus, abnormal

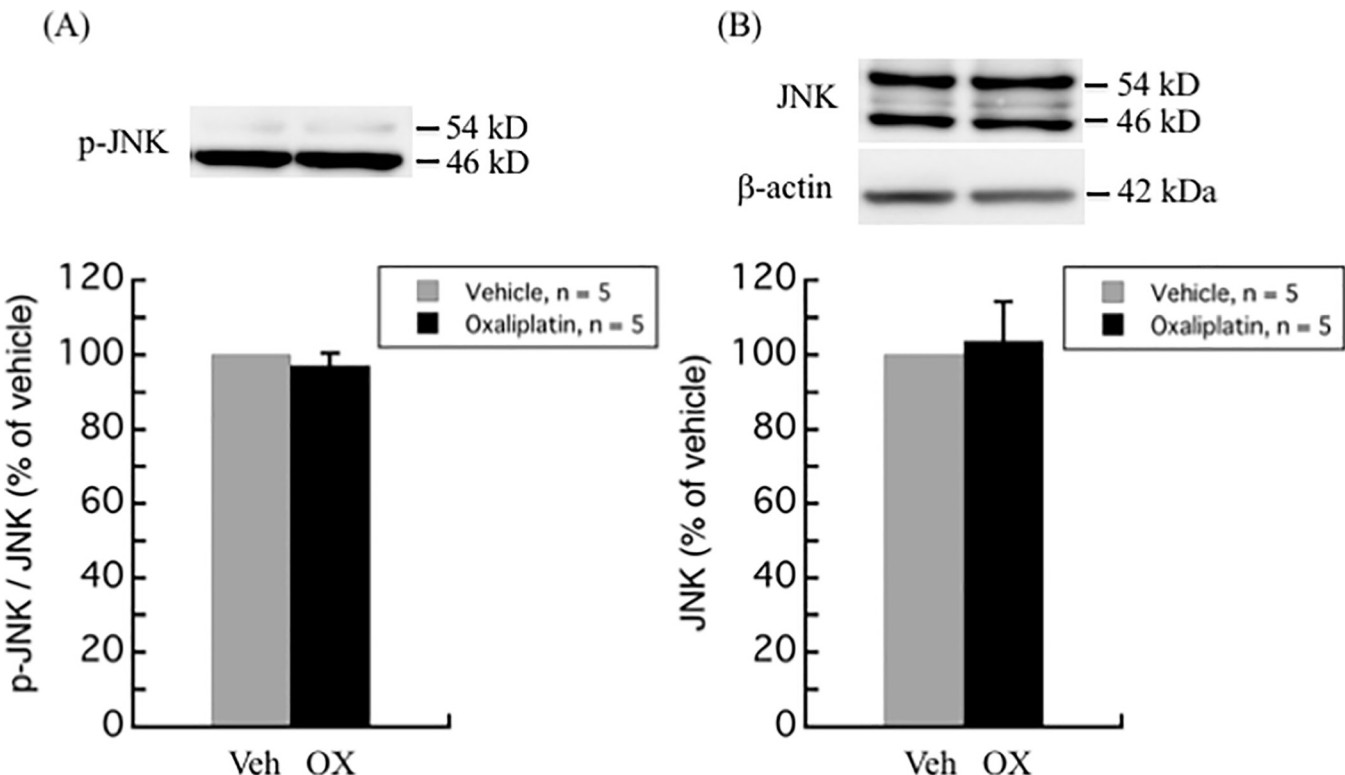

**Fig 4. JNK phosphorylation and protein levels in rat DRG.** There was no difference in JNK phosphorylation and protein levels between oxaliplatin and vehicle treatment groups. Comparisons between two groups of the blots were performed by Student's t-test. All data are calculated as mean ± SD of 5 animals.

$Na^+$ channel and/or $K^+$ channel function has been highlighted as a possible mechanism of oxaliplatin-induced neuropathy. However, focusing on $Na^+$ channel subtypes, $Na_V1.7$, $Na_V1.8$, and $Na_V1.9$, expressed in the DRG, conditional knockout mice established using the $Na_V1.7^{Advill}$ line, which eliminates $Na_V1.7$ expression in all the DRG neurons, and the $Na_V1.7^{Wnt1}$ line, which lacks $Na_V1.7$ expression in the DRG and sympathetic ganglion neurons, both $Na_V1.7^{Advill}$ and $Na_V1.7^{Wnt1}$ mice developed mechanical and cold allodynia normally following oxaliplatin treatment [32]. In addition, global deletion of $Na_V1.3$, $Na_V1.8$, or $Na_V1.9$ also did not attenuate either mechanical or cold allodynia in oxaliplatin-induced neuropathy. Furthermore, in our study, the expression levels of $Na_V1.7$, $Na_V1.8$, and $Na_V1.9$ were not altered in the DRG of oxaliplatin treated rats compared to non-treated rats (S1 Fig and S2 Fig). These findings suggest that the expression of the voltage-dependent $Na^+$ channel subtypes $Na_V1.3$, $Na_V1.7$, $Na_V1.8$, and $Na_V1.9$ is not required for the development of oxaliplatin-induced neuropathy.

So, how does abnormal $Na^+$ current lead to the development of nerve hyperexcitability? Firstly, $Na^+$ channel subtypes other than $Na_V1.3$, $Na_V1.7$, $Na_V1.8$, and $Na_V1.9$ could contribute to oxaliplatin-induced nerve hyperexcitability. Indeed, a recent study revealed that the expression of $Na_V1.6$ was dramatically increased in the DRG in oxaliplatin-induced CIPN model rats [33]. Furthermore, the agomir of miR-30b, a microRNA implicated in neuropathic pain, cancer, and neurodegenerative diseases, can downregulate $Na_V1.6$ and alleviate oxaliplatin-induced mechanical allodynia and cold hypersensitivity [33]. Secondly, the cytokines and chemokines associated with altering intracellular signaling could modify $Na^+$ current, which is conducive to nerve hyperexcitability. Recently accumulated evidence has shown that oxaliplatin treatment increases pro- and anti-inflammatory cytokines and chemokines [34–40].

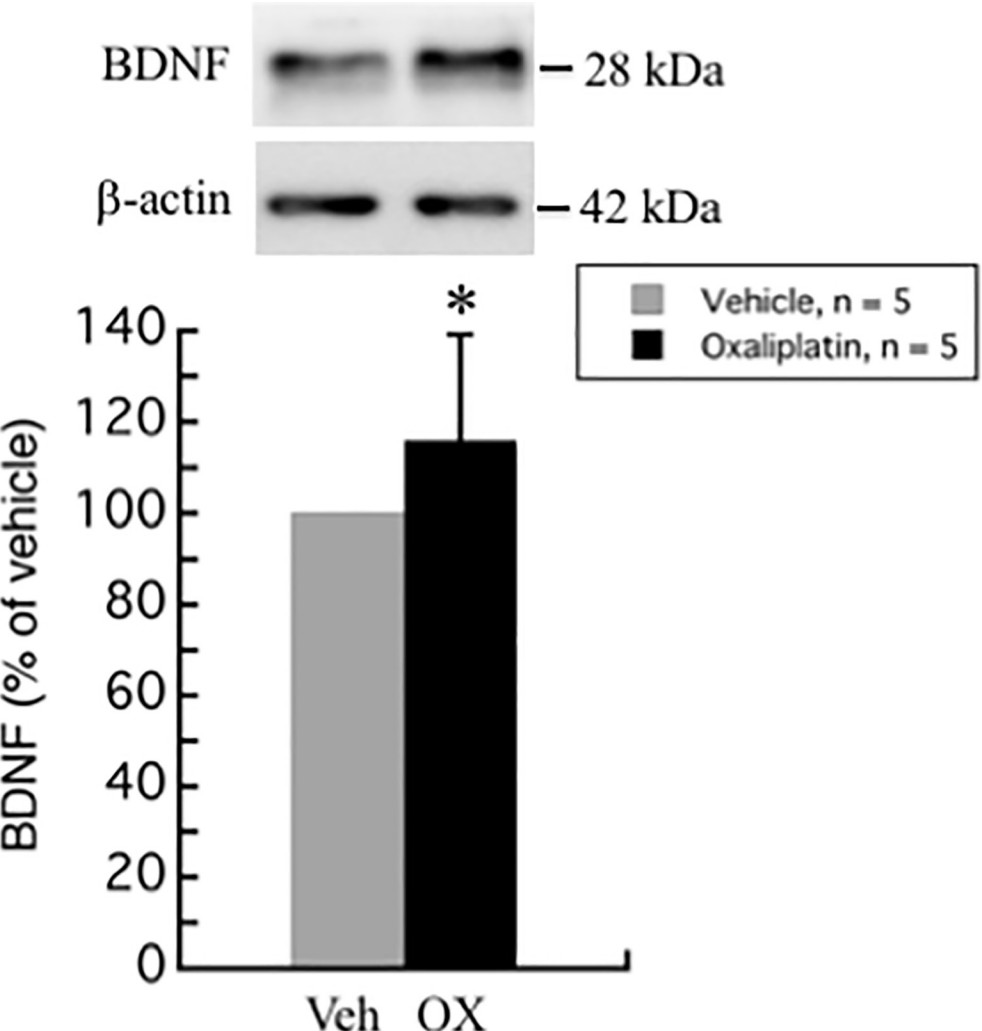

**Fig 5. Upregulation of BDNF by oxaliplatin in rat DRG.** The expression of BDNF was significantly increased in DRG of oxaliplatin treated rats. Comparisons between two groups of the blots were performed by Student's t-test. All data are calculated as mean ± SD of 5 animals. * $P < 0.05$, compared to the vehicle.

Oxaliplatin injection enhanced the mRNA levels of cytokines including tissue necrosis factor-α (TNF-α) and interleukin-1β (IL-1β), and chemokines including monocyte chemoattractant protein-1 (MCP-1, also referred to as C-C chemokine ligand (CCL) 2) and monocyte inflammatory protein-1 (MIP-1α, also referred to as CCL3) in the spinal dorsal horn [34]. Melatonin attenuates pain hypersensitivity by inhibition of TNF-α in oxaliplatin-induced neuropathy [33]. CCL2 and its receptor CCR2 have also been shown to be increased in the DRG after oxaliplatin administration, in parallel with the development of mechanical hypersensitivity [35, 36]. Wang et al. have reported that oxaliplatin treatment up-regulates NF-κB and induces neuronal hyperexcitability in DRG [37]. This neuronal hyperexcitability was inhibited by NF-κB inhibitors. Oxaliplatin-induced pain has also been shown to be accompanied with the upregulation of PI3K-mTOR and mTOR-mediated signals as well as IL-1β, IL-6, and TNF-α in DRG. As PI3K or mTOR signal was inhibited, mechanical and cold hypersensitivity were attenuated in oxaliplatin treated rats, and the levels of proinflammatory cytokines also decreased [38]. The upregulation of pro-inflammatory cytokines and membrane pro-inflammatory cytokine

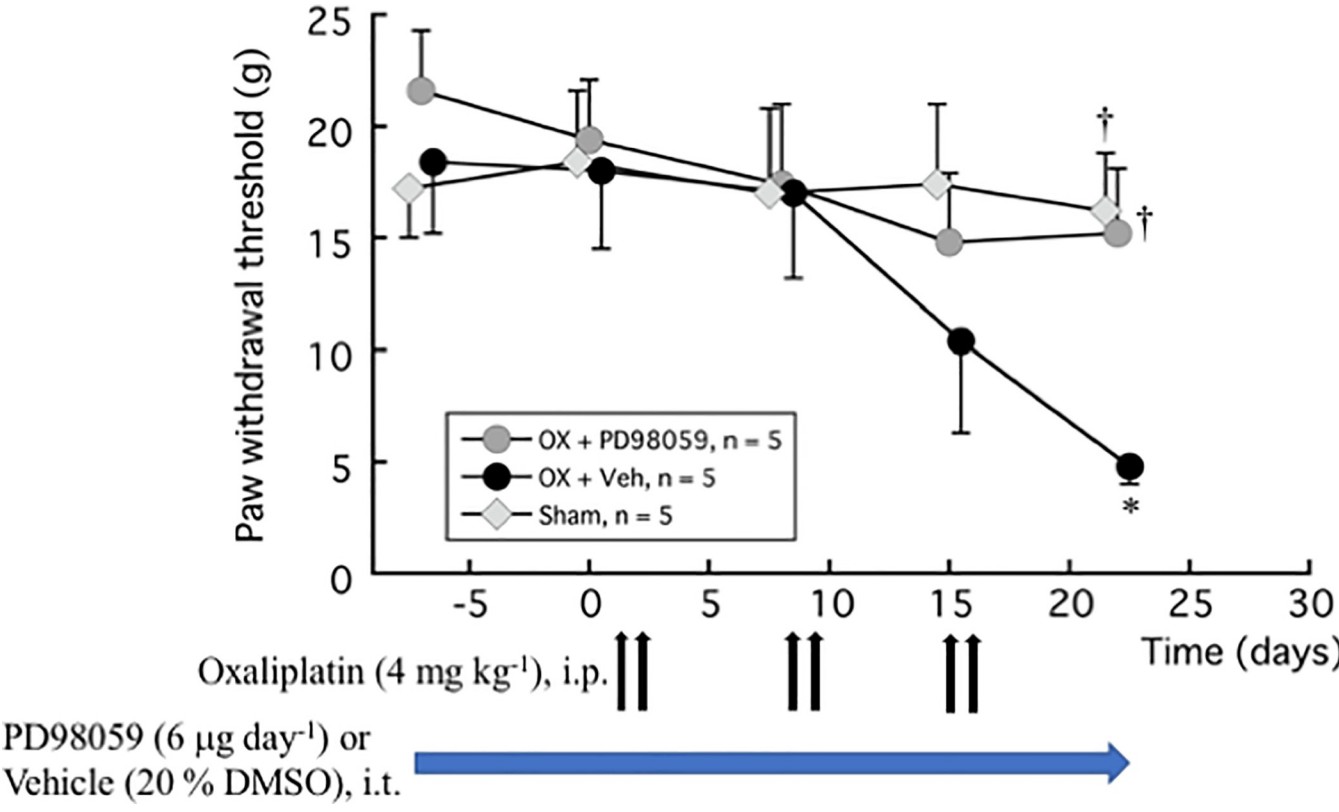

**Fig 6. Inhibition of oxaliplatin-induced mechanical allodynia by ERK inhibitor PD98059.** PD98059 (6 μg day$^{-1}$) or vehicle (20% DMSO) was injected i.t. for 4 weeks using an osmotic pressure pump. One week after the pump placement, oxaliplatin (4 mg kg$^{-1}$) or vehicle (5% glucose) was injected i.p. twice a week for 3 weeks. von Frey test was done before the pump placement, and before and 1 week after each oxaliplatin or vehicle treatment (days 1, 2, 8, 9, 15, and 16). We confirmed the incidence of mechanical allodynia on day 22 (day 28 from pump placement). The hindpaw data within each group were analyzed using one-way repeated measures ANOVA followed by Bonferroni post hoc analysis. Welch's test was used to compare between groups. All data are calculated as mean ± SEM of 5 animals. * $P < 0.05$, compared with Time -7 days (baseline). $^{†}$ $P < 0.05$, compared with OX + Veh at the same time.

receptors in the midbrain periaqueductal gray, which has an inhibitory or excitatory control on pain transmission via the rostral ventromedial medulla, projecting to the spinal dorsal horn, of oxaliplatin treated rats is likely to impair the descending inhibitory pathways in regulation of pain transmission and thereby, contribute to the development of neuropathic pain after the administration of chemotherapeutic oxaliplatin [39]. These reports suggest that the mechanism of development of oxaliplatin-induced neuropathy resembles inflammatory pain. Furthermore, Huang et al. and Liu et al. have reported an increase in the levels of TNF-α, NF-κB, and phosphorylation of ERK in the spinal cord and DRG of an oxaliplatin-induced peripheral neuropathy rat model and a lumber disk herniation rat model [36, 40]. Previous studies have demonstrated that TNF-α enhances TTX-R Na$^+$ currents via TNFR1 and the p38 pathway in the cultured DRG neurons within 1 minute of the onset of TNF-α application (peak effect within 3–5 minutes) [41], as well as via the p38 pathway in the uninjured DRG neurons after L5-ventral root transection (VRT) *in vivo* [42]. The phosphorylated ERK1 lowers the activation threshold, making it easier to open Na$_V$1.7 channel in response to weak stimuli [43]. We have also reported that veratridine-induced $^{22}$Na$^+$ influx was inhibited by the inhibitors of ERK and p38, indicating that the basal constitutive activities of ERK and p38 may prime Na$_V$1.7 to open [44]. These findings suggest that oxaliplatin-induced neuroinflammation and

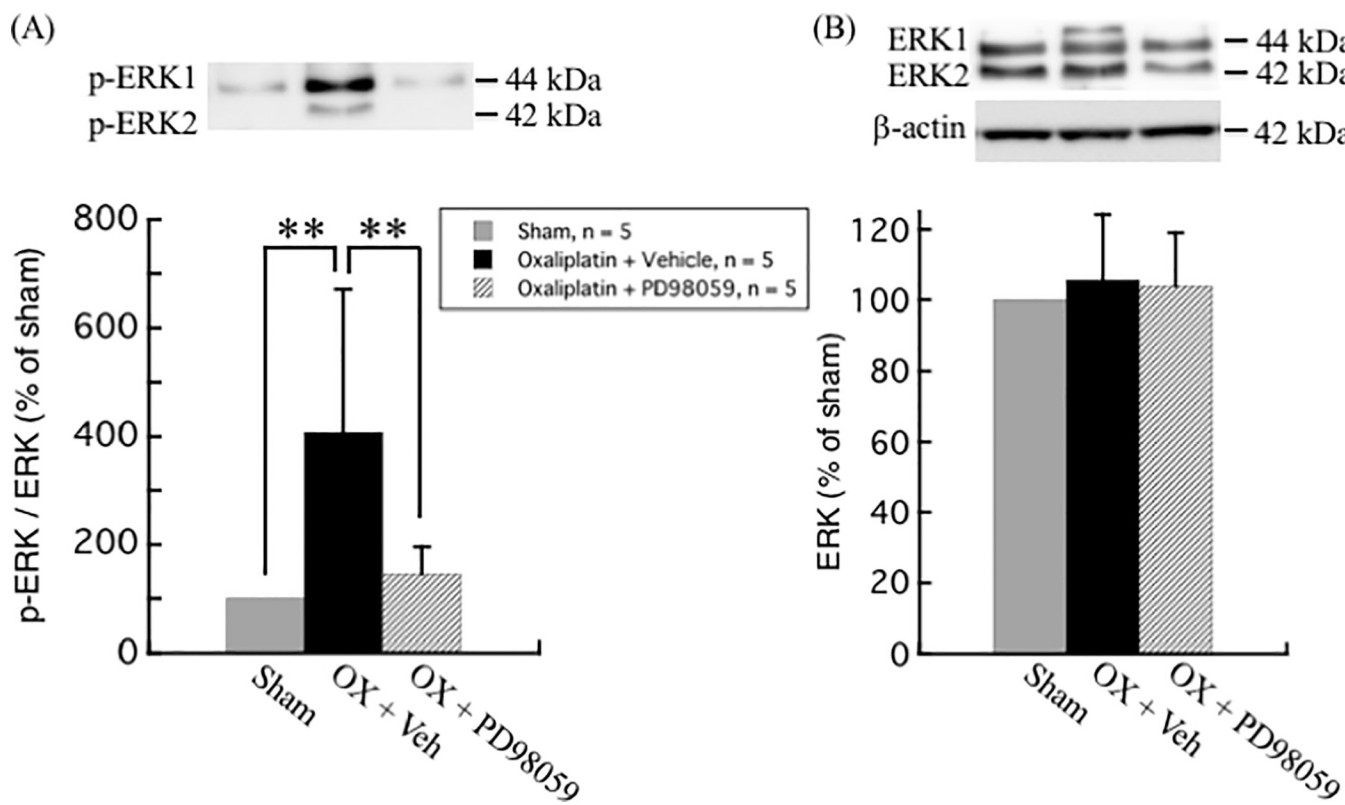

**Fig 7. Effect of ERK inhibitor PD98059 on oxaliplatin-induced upregulation of ERK phosphorylation in rat DRG.** (A) The ratio of p-ERK to ERK expression was significantly increased in DRG of oxaliplatin treated rats, which was inhibited by PD98059. (B) No difference was observed in the protein level of ERK among sham, oxaliplatin + vehicle (20% DMSO), and oxaliplatin + PD98059 treatment groups. Comparisons between two groups of the blots were performed by Student's t-test. All data are calculated as mean ± SD of 5 animals. ** $P < 0.01$, compared with OX + Veh.

inflammatory mediators would evolve abnormal $Na^+$ channel currents via MAPK including ERK phosphorylation, and thus, lead to the development of neuropathy.

## Conclusions

Oxaliplatin administration induces chronic mechanical allodynia in rats. The phosphorylation of ERK is upregulated in the DRG of oxaliplatin-induced neuropathic pain rat model, whereas other MAPKs, p38, and JNK are not altered. ERK inhibitor impedes mechanical allodynia by inhibiting oxaliplatin-induced upregulation of ERK phosphorylation. Thus, the findings from the present study may provide a better understanding of the intracellular molecular alterations in the development of oxaliplatin-induced neuropathic pain and help in designing effective therapeutics.

## Supporting information

**S1 Fig. The protein levels of $Na_V1.7$, $Na_V1.8$, and $Na_V1.9$ in oxaliplatin-treated rat DRG.** Typical western blots of $Na_V1.7$, $Na_V1.8$, and $Na_V1.9$ are shown. These images suggested that there was no difference in $Na_V1.7$, $Na_V1.8$, and $Na_V1.9$ protein levels between oxaliplatin and vehicle treatment groups.
(TIFF)

**S2 Fig. The mRNA levels of Na$_V$1.7, Na$_V$1.8, and Na$_V$1.9 in oxaliplatin-treated rat DRG.**
Typical polymerase chain reaction (PCR) gel images of Na$_V$1.7, Na$_V$1.8, and Na$_V$1.9 are
shown. These images suggested that there was no difference in Na$_V$1.7, Na$_V$1.8, and Na$_V$1.9
mRNA expression levels between oxaliplatin and vehicle treatment groups.
(TIFF)

**S3 Fig. Expanded views of western blots and PCR used in Figs 2–5, 7, S1 and S2.**
(TIFF)

## Acknowledgments

This study is attributed to the Department of Anesthesiology, Faculty of Medicine, University
of Miyazaki. The authors would like to thank Noriko Hidaka, Mio Kurogi, and Toshiko Wata-
nabe for their technical and secretarial assistance in this study. The authors would like to
thank Editage (www.editage.jp) for English language editing.

## Author Contributions

**Conceptualization:** Toyoaki Maruta, Takayuki Nemoto, Isao Tsuneyoshi.

**Data curation:** Toyoaki Maruta, Takayuki Nemoto.

**Formal analysis:** Toyoaki Maruta, Takayuki Nemoto, Tetsuro Shirasaka, Toshihiko Yanagita,
Ryu Takeya, Isao Tsuneyoshi.

**Funding acquisition:** Toyoaki Maruta.

**Investigation:** Toyoaki Maruta, Takayuki Nemoto, Koutaro Hidaka, Tomohiro Koshida.

**Project administration:** Toyoaki Maruta.

**Writing – original draft:** Toyoaki Maruta, Takayuki Nemoto.

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
