## [Decision Letter · Decision Letter 0]

22 Aug 2019

PONE-D-19-17881

Upregulation of ERK phosphorylation in rat dorsal root ganglion neurons contributes to oxaliplatin-induced chronic neuropathic pain

PLOS ONE

Dear Dr. MARUTA,

Thank you for submitting your manuscript to PLOS ONE. After careful consideration, we feel that it has merit but does not fully meet PLOS ONE’s publication criteria as it currently stands. Therefore, we invite you to submit a revised version of the manuscript that addresses the points raised during the review process.

Please address the reviewers' points, especially those criticizing the statistical analyses.

We would appreciate receiving your revised manuscript by Oct 06 2019 11:59PM. To enhance the reproducibility of your results, we recommend that if applicable you deposit your laboratory protocols in protocols.io, where a protocol can be assigned its own identifier (DOI) such that it can be cited independently in the future. For instructions see: http://journals.plos.org/plosone/s/submission-guidelines#loc-laboratory-protocols

We look forward to receiving your revised manuscript.

Kind regards,

Ferenc Gallyas, Jr., Ph.D., D.Sc.

Academic Editor

PLOS ONE

Journal Requirements:

2. At this time, we request that you  please report additional details in your Methods section regarding animal care, as per our editorial guidelines:

a) Please describe any steps taken to minimize animal suffering and distress, such as by administering analgesics,

b) Please include the method of sacrifice and

c) Please describe the post-operative care received by the animals, including the frequency of monitoring and the criteria used to assess animal health and well-being.

Thank you for your attention to these requests.

Reviewers' comments:

Reviewer's Responses to Questions

**Comments to the Author**

1. Is the manuscript technically sound, and do the data support the conclusions?

Reviewer #1: Partly

Reviewer #2: Partly

2. Has the statistical analysis been performed appropriately and rigorously? 

Reviewer #1: No

Reviewer #2: Yes

3. Have the authors made all data underlying the findings in their manuscript fully available?

Reviewer #1: Yes

Reviewer #2: Yes

4. Is the manuscript presented in an intelligible fashion and written in standard English?

Reviewer #1: Yes

Reviewer #2: Yes

5. Review Comments to the Author

Reviewer #1: In the current manuscript, the authors investigate the factors underlying the neuropathy development following oxaliplatin treatment in rats. This phenotype is seen after 3-4 weeks of treatment. The authors propose that upregulation of ERK phosphorylation in rat DRG neurons is responsible for generating this pain phenotype. They use ERK inhibition in combination with the evaluation of mechanical pain and western blot analysis to show that this mechanism may be responsible for the oxaliplatin-induced neuropathic pain. The data are conciseness and supportive, the study is well written and the conclusions are broadly reasonable. To improve the paper the authors should consider the following points:

1. The authors need to increase the sample size to ascertain that this time course of the incidence of mechanical allodynia in oxaliplatin group respect to vehicle group is not related to the small sample size -i.e. n=5 per group. The control group show a high variability These results will provide crucial information to the interpretation of the current data set (Fig1).

2. The authors should include more information that clarifies and justifies their statistical analyses. Specify the statistical methods performed in the figure legends and include the significance values in manuscript text. Please also provide all vehicle group values (mean +/- SD) which are used to relative the oxaliplatin group values, and F and P-values for each individual factor for all statistic performed.

3. For the ERK inhibition experiments on oxaliplatin-induced neuropathic pain, important further controls include sham rats that receive vehicle and ERK inhibitor in behavioral test and western blot. While this represents significant additional experiments, it would help interpret the existing main findings.

4. There have been previous reports of acute oxaliplatin administration already contributes to mechanical hyperalgesia (peripheral neuropathic pain) in DRG involving different mechanisms (Huang et al., 2018; Illias et al., 2018) - perhaps this should be integrated in their discussion.

Reviewer #2: This paper studied the factors contributing to the induction of chemotherapy-induced peripheral neuropathy (CIPN), using the platinum based chemotherapeutic agent, oxaliplatin. The authors used a rat model of oxaliplatin-induced neuropathic pain to evaluate the role of different mitogen-activated protein kinase (MAPK) pathways in the dorsal root ganglia of the lumbar spinal cord segment. The authors were specifically interested in studying the expression of ERK in the DRG, and the effect oxaliplatin treatment had on ERK phosphorylation in relation to the onset of mechanical allodynia. The results from these experiments report that oxaliplatin induces phosphorylation of EDK, and that delivering an ERK inhibitor inhibited oxaliplatin-induced ERK phosphorylation and blocked the onset of mechanical allodynia.

This paper provides great insight on the pathology underlying the development of cancer induced neuropathic pain. A lot of research in the pain field has focused on other models of chronic pain, such as diabetic neuropathy and peripheral nerve injury, but mechanisms underlying cancer therapeutic induced neuropathic pain are still not understood. I believe that these findings would contribute to furthering our knowledge on the pathology driving chronic pain caused by anti-cancer agents. There are questions and comments that should be addressed, before moving forward.

Abstract/Introduction:

• Lines 29-30: The statement "we aimed to understand the factors involved in the development and maintenance of chronic neuropathy elicited by oxaliplatin treatment" is misleading. The pathology underlying, both, the development and maintenance of chronic pain is a complex, multi-phase process involving changes in peripheral and central nervous system. The design of this study focused on the mechanism driving the onset of mechanical allodynia in relation to the effect oxaliplatin treatment had on the expression of p-ERK within the DRG, rather than the process of transitioning from acute to chronic pain. Clarifying the aim will strengthen the conclusions made by the authors, and help readers better interpret the data presented in each experiment. This statement is also made again in lines 68-70.

• Lines 49-60: Background information is introduced only on sodium ion channels and TRP channels, indicating their role in temperature hypersensitivity. A brief overview of current literature on mechanoreceptors in pain models should be introduced to reinforce that the author's are specifically studying mechanical allodynia because it's pathophysiology is poorly understood in this model of chronic pain.

Methods:

• Lines 87-112: Why was there a difference in duration of treatment between experiments 1 and 2? Do you have data comparing p-ERK/ERK expression in animals receiving treatment for 3 weeks versus animals receiving treatment for 4 weeks?

• Lines 107-108: The sham procedure is unclear. Did sham animals receive any i.p. or i.t. treatment, or were they naïve?

Results:

• The figure legends indicate values are expressed as mean +/- SD, but the values aren't reported in the results sections, figure legends, or on the graphs.

• The number of animals assigned to each treatment group should be reported on all the graphs.

• Lines 228-241: The mechanical allodynia measurements from the sham group need to be included in Figure 6 and incorporated into the statistical analysis. The sham behavioral data should be presented in this figure in order compare all three groups in Figure 7.

Discussion:

• Lines 285-289: The authors expand on the findings from previous studies concerning the involvement of voltage dependent sodium channel subtypes in the pathophysiology of oxaliplatin-induced neuropathy. The author's findings from the experiments on Na1.7, Na1.8, and Na1.9 expression in the DRG are consistent with the literature, but recent studies have reported that Na1.6 expression in the DRG contributes to the onset of oxaplatin induced neuropathic pain. The reports characterize Na1.6 in the DRG for its role in pain behavior as well as having abnormal spontaneous neuronal activity following nerve injury. Evidence from these studies show that directly targeting Na1.6 in the DRG alleviates oxaplatin induced mechanical allodynia and cold hypersensitivity. These studies should be reviewed and interpreted in the context of this study in the final paragraph of the discussion.

Overall, I thought this was a very well executed study that will contribute to the CIPN research community. After these points are addressed, I believe the manuscript will be ready to move forward in the review process as it satisfies the PLOS ONE criteria for publication.

6. PLOS authors have the option to publish the peer review history of their article (what does this mean?). If published, this will include your full peer review and any attached files.

Reviewer #1: No

Reviewer #2: No

---

## [Author Response · Author response to Decision Letter 0]

2 Oct 2019

Dear Reviewer #1 (Manuscript ID: PONE-D-19-17881; Title: Upregulation of ERK phosphorylation…)

Thank you for your valuable comments on our paper. In response to your comments, we have extensively revised our paper, as shown below in a point-by-point manner.

Points 1 & 2. Reviewer #1 pointed out that the sample size of the behavioral experiments, especially in the vehicle group, were small and that more information about the data and statistical analysis should be provided. 

Considering this, we reviewed our statistical analysis. First, we must apologize for a miscalculation of SEM from SD. SEM = SD / √n (n = number of sample size), so SD / √5 was correct, but we miscalculated by using SD / √3. We have now revised all the SEM data. 

Second, in the comparisons between two groups of the hind paw data from the behavioral experiments, the F test shows P < 0.05, thus in this revised version of the manuscript we have used Welch’s test, not Student’s t-test. In addition, we have added the statistical methods in the figure legends, and we have provided the P-value in manuscript text when there are significant differences. 

Regarding sample size, we do not think that we need more samples. We agree that if the sample size of vehicle group were increased, the SD and SEM would likely be narrower (because variability would become low). However, we think that as there were already statistical significances, we should minimize the number of animals used in each experiment.

Reviewer #1 stated that we should provide all vehicle group values (mean ± SD) which are used to relative the oxaliplatin group values. In our present study, western blot data were calculated as percentages of the control blot density, which was expressed as 100% (control meant vehicle or sham group in each experiment) on each membrane. The blot data were presented as means of these percentages ± SD. We consider this to be a common way to present expression analysis from western blots. For example, in figure 2 (A), the p-ERK level of oxaliplatin group was 447.6 ± 273.6% as a percentage of the vehicle, whereas the densities of protein blots quantified by ImageJ in the vehicle vs. oxaliplatin were 0.31 ± 0.19 vs. 0.96 ± 0.37, respectively. However, we consider that the calculation of densities is inaccurate in our present study, because we did not use an internal standard (e.g. the same DRG sample) on each membrane. Thus, we could not normalize the densities of the blots on each membrane.

Point 3. Reviewer #1 pointed out that for the ERK inhibition experiments, sham rats receiving vehicle (5% glucose) i.p. and ERK inhibitor i.t. (Veh + PD98059) should be used, because it would help interpret our findings. 

Of course, we also considered whether a Veh + PD98059 group was needed or not when designing the experiments. We researched previous studies in which similar ERK inhibition experiments were performed. Some studies (marked below with*) did not use a Veh + PD98059 group. When there was a Veh + inhibitor treatment group, the ERK inhibitor did not affect the behavioral test (marked below with**). Furthermore, in our present study, Figure 6 & 7 show that the ERK inhibitor clearly inhibits ERK phosphorylation in DRG and oxaliplatin-induced mechanical allodynia. Therefore, we decided not to use a Veh + PD98059 group.

*(1) Cao Y, Li K, Fu KY, Xie QF, Chiang CY, Sessle BJ. Central sensitization and MAPKs are involved in occlusal interference-induced facial pain in rats. J Pain. 2013;14:793-807. (2) Wang XW, Li TT, Zhao J, Mao-Ying QL, Zhang H, Hu S, Li Q, Mi WL, Wu GC, Zhang YQ, Wang YQ. Extracellular signal-regulated kinase activation in spinal astrocytes and microglia contributes to cancer-induced bone pain in rats. Neuroscience. 2012;217:172-181. (3) Yoon SY, Kwon SG, Kim YH, Yeo JH, Ko HG, Roh DH, Kaang BK, Beitz AJ, Lee JH, Oh SB. A critical role of spinal Shank2 proteins in NMDA-induced pain hypersensitivity. Mol Pain. 2017;13:1744806916688902.

**(1) Sanna MD, Mello T, Ghelardini C, Galeotti N. Inhibition of spinal ERK1/2-c-JUN signaling pathway counteracts the development of low doses morphine-induced hyperalgesia. Eur J Pharmacol. 2015;764:271-277. (2) Xing F, Kong C, Bai L, Qian J, Yuan J, Li Z, Zhang W, Xu JT. CXCL12/CXCR4 signaling mediated ERK1/2 activation in spinal cord contributes to the pathogenesis of postsurgical pain in rats. Mol Pain. 2017;13:1744806917718753.

Point 4. Reviewer #1 pointed out that the reports of Huang et al. 2018 and Illias et al. 2018 should be integrated into the discussion. Therefore, we added a discussion of these reports (lines 331-333 and lines 347-349). We also added these reports to reference list (35 and 36). 

We felt that Reviewer #1 emphasized acute oxaliplatin administration. There are various oxaliplatin administration methods that produce oxaliplatin-induced mechanical allodynia. In some methods, single or short administration of oxaliplatin produces mechanical allodynia in the early phase, such as thermal hyperalgesia. We chose a method similar to the clinical administration and mechanical allodynia onset for humans. In our present report, we did not refer to the variation of mechanical allodynia due to the differences in the oxaliplatin administration methods. 

Dear Reviewer #2 (Manuscript ID: PONE-D-19-17881; Title: Upregulation of ERK phosphorylation…)

Thank you for your valuable comments on our paper. In response to your comments, we have extensively revised our paper, as shown below point-by-point manner.

Abstract/Introduction:

 Lines 29-30 & 68-70: We agree your suggestion and revise the sentence from “development and maintenance” to “generation” (line 30 and line 78).

 Lines 49-60: Reviewer #2 pointed out that mechanoreceptors in pain model should be introduced to reinforce that we studied oxaliplatin-induced mechanical allodynia, not thermal hyperalgesia. However, neuropathic pain is caused not only by mechanoreceptors, so we added some sentences about the potential mechanisms of neuropathic pain (lines 57-64) in the introduction and added the following reports to the references (3. Baron R, Binder A, Wasner G. Neuropathic pain: diagnosis, pathophysiological mechanisms, and treatment. Lancet Neurol. 2010;9:807-819. and 4. St John Smith E. Advances in understanding nociception and neuropathic pain. J Neurol. 2018;265:231-238.).

 Lines 87-112: Reviewer #2 pointed out the difference in treatment duration between experiment 1 and 2. We used an ALZET osmotic pressure pump for continuous injection of PD98059. We could find an ideal pump for our experiments (size, infusion rate, and infusion duration), however, this pump can only continue to infuse the drug for 4 weeks. Therefore, we had to choose the data in animals receiving oxaliplatin treatment for 3 weeks, in which the paw withdrawal thresholds were significantly lower than vehicle treatment. 

 Lines 107-108: Reviewer#2 pointed out that the sham procedure is unclear. However, we already described “The sham-operated (Sham) mice underwent a similar surgical procedure except for pump placement and drug treatments” in Method section (line 119-120). Therefore, the sham-operated mice (Sham) did not receive any i.p. or i.t. treatment.

Results:

 We changed the sentences from “Values are expressed” to “All data are calculated” in each figure legend. We also added the values in manuscripts text when there are significant differences.

 The number of animals assigned to each treatment group are already described in the figure legends, but not reported on the graphs. Therefore, we added the number on each graph.

 Lines 228-241: We did not show the sham group in Figure 6, because the hind paw data of the sham group was not altered compared with the OX + PD98059 group and we were afraid that Figure 6 would become too busy and difficult to interpret. However, we agree with the reviewer's comment and have added the sham group in Figure 6.

Discussion:

 Lines 285-289: We found a recent report that the expression of NaV1.6 in DRG was increased in oxaliplatin-induced CIPN model rats. We review this study in the discussion (lines 317-324) and have added this report to the references (33. Li L, Shao J, Wang J, Liu Y, Zhang Y, Zhang M, Zhang J, Ren X, Su S, Li Y, Cao J, Zang W. MiR-30b-5p attenuates oxaliplatin-induced peripheral neuropathic pain through the voltage-gated sodium channel NaV1.6 in rats. Neuropharmacology. 2019;153:111-120.).

---

## [Decision Letter · Decision Letter 1]

29 Oct 2019

PONE-D-19-17881R1

Upregulation of ERK phosphorylation in rat dorsal root ganglion neurons contributes to oxaliplatin-induced chronic neuropathic pain

PLOS ONE

Dear Dr. MARUTA,

Thank you for submitting your manuscript to PLOS ONE. After careful consideration, we feel that it has merit but does not fully meet PLOS ONE’s publication criteria as it currently stands. Therefore, we invite you to submit a revised version of the manuscript that addresses the points raised during the review process.

Please address the statistical analysis issues raised by reviewer#1.

We would appreciate receiving your revised manuscript by Dec 13 2019 11:59PM. To enhance the reproducibility of your results, we recommend that if applicable you deposit your laboratory protocols in protocols.io, where a protocol can be assigned its own identifier (DOI) such that it can be cited independently in the future. For instructions see: http://journals.plos.org/plosone/s/submission-guidelines#loc-laboratory-protocols

We look forward to receiving your revised manuscript.

Kind regards,

Ferenc Gallyas, Jr., Ph.D., D.Sc.

Academic Editor

PLOS ONE

Reviewers' comments:

Reviewer's Responses to Questions

**Comments to the Author**

1. If the authors have adequately addressed your comments raised in a previous round of review and you feel that this manuscript is now acceptable for publication, you may indicate that here to bypass the “Comments to the Author” section, enter your conflict of interest statement in the “Confidential to Editor” section, and submit your "Accept" recommendation.

Reviewer #1: (No Response)

Reviewer #2: All comments have been addressed

2. Is the manuscript technically sound, and do the data support the conclusions?

Reviewer #1: Partly

Reviewer #2: Yes

3. Has the statistical analysis been performed appropriately and rigorously? 

Reviewer #1: No

Reviewer #2: Yes

4. Have the authors made all data underlying the findings in their manuscript fully available?

Reviewer #1: Yes

Reviewer #2: Yes

5. Is the manuscript presented in an intelligible fashion and written in standard English?

Reviewer #1: Yes

Reviewer #2: Yes

6. Review Comments to the Author

Reviewer #1: Comments to authors:

In the revised manuscript, the authors have adequately revised the manuscript and addressed important questions regarding the experimental results and analysis. But if possible, I would still like to address some point that were raised in the review:

1) In the behavioral experiments, in the comparisons between time points of the hind paw data is recommendable to perform a repeated measures ANOVA.

2) In western blot experiments, I agree with the authors that calculating data as percentages of control blot density (expressed as 100%) is a common way of representation. What I pointed out in the previous review is to express somehow the variance within the control group, since although it is considered 100%, it is an average of measures with SD. The protein density data quantified in ImageJ for experimental and control groups would support the levels of significance expressed in the figures (above suggested supplementary table). Also, because I understand that the statistical analysis has been carried out with these data. According to the authors, I consider that methodologically the most adequate quantification has not been performed due to the absence of internal standard. However, if it is possible to normalize the density of protein blots of each membrane with respect to the background of each of them. Have you been considered in the analysis?

Reviewer #2: (No Response)

7. PLOS authors have the option to publish the peer review history of their article (what does this mean?). If published, this will include your full peer review and any attached files.

Reviewer #1: No

Reviewer #2: Yes: Zach LaMacchia

---

## [Author Response · Author response to Decision Letter 1]

4 Nov 2019

Dear Reviewer #1 (Manuscript ID: PONE-D-19-17881; Title: Upregulation of ERK phosphorylation…)

Thank you for your valuable comments on our paper. In response to your comments, we have extensively revised our paper, as shown below in a point-by-point manner.

Points 1. Reviewer #1 pointed out that in the comparisons between time points of the hind paw data is recommendable to perform a repeated measures ANOVA. 

So, we used one-way repeated ANOVA. OX group in Figure 1 and OX + Vehicle group in Figure 6 were significantly changed according to time (p = 0.0003 and p = 0.0039, respectively). We revised the sentences in statistical methods and Figure legends (lines 175, 195, and 260).

Point 2. Reviewer #1 pointed out that to express somehow the variance within the control group, since although it is considered 100%, it is an average of measures with SD; and if it is possible to normalize the density of protein blots of each membrane with respect to the background of each of them. 

Of course, the density of protein blots in control group must have the variance. We tried to use the background density as an internal standard. For example, in figure 2 (A), the p-ERK level of oxaliplatin group was 447.6 ± 273.6% as a percentage of the vehicle, whereas the densities of protein blots quantified by ImageJ in the vehicle vs. oxaliplatin were 0.31 ± 0.19 vs. 0.96 ± 0.37 (p = 0.0006), respectively; and when using the background density as an internal standard, the densities of protein blots in the vehicle vs. oxaliplatin were 0.46 ± 0.32 vs. 1.39 ± 0.65 (p = 0.003), respectively. This way to use the background density as an internal standard seems good and we are grateful for Reviewer #1’s valuable advice. However, we have some concerns that this way might depend on how to set background and we do not know whether there is consensus in the use of the background density as an internal standard. We consider that using same sample as an internal standard on each membrane is better, if the variance of control is needed to be expressed. So, in this study, we would like to keep to express the calculating data as percentages of control blot density (expressed as 100%).

---

## [Editor Report · Decision Letter 2]

8 Nov 2019

Upregulation of ERK phosphorylation in rat dorsal root ganglion neurons contributes to oxaliplatin-induced chronic neuropathic pain

PONE-D-19-17881R2

Dear Dr. MARUTA,

We are pleased to inform you that your manuscript has been judged scientifically suitable for publication and will be formally accepted for publication once it complies with all outstanding technical requirements.

With kind regards,

Ferenc Gallyas, Jr., Ph.D., D.Sc.

Section Editor

PLOS ONE
---

## [Editor Report · Acceptance letter]

13 Nov 2019

PONE-D-19-17881R2 

Upregulation of ERK phosphorylation in rat dorsal root ganglion neurons contributes to oxaliplatin-induced chronic neuropathic pain 

Dear Dr. MARUTA:

I am pleased to inform you that your manuscript has been deemed suitable for publication in PLOS ONE. Congratulations! Your manuscript is now with our production department. 

With kind regards,

on behalf of

Dr. Ferenc Gallyas, Jr. 

Section Editor

PLOS ONE